# Swarm learning with weak supervision enables automatic breast cancer detection in magnetic resonance imaging

## Abstract

**Background** Over the next 5 years, new breast cancer screening guidelines recommending magnetic resonance imaging (MRI) for certain patients will significantly increase the volume of imaging data to be analyzed. While this increase poses challenges for radiologists, artificial intelligence (AI) offers potential solutions to manage this workload. However, the development of AI models is often hindered by manual annotation requirements and strict data-sharing regulations between institutions.

**Methods** In this study, we present an integrated pipeline combining weakly supervised learning—reducing the need for detailed annotations—with local AI model training via swarm learning (SL), which circumvents centralized data sharing. We utilized three datasets comprising 1372 female bilateral breast MRI exams from institutions in three countries: the United States (US), Switzerland, and the United Kingdom (UK) to train models. These models were then validated on two external datasets consisting of 649 bilateral breast MRI exams from Germany and Greece.

**Results** Upon systematically benchmarking various weakly supervised two-dimensional (2D) and three-dimensional (3D) deep learning (DL) methods, we find that the 3D-ResNet-101 demonstrates superior performance. By implementing a real-world SL setup across three international centers, we observe that these collaboratively trained models outperform those trained locally. Even with a smaller dataset, we demonstrate the practical feasibility of deploying SL internationally with on-site data processing, addressing challenges such as data privacy and annotation variability.

**Conclusions** Combining weakly supervised learning with SL enhances inter-institutional collaboration, improving the utility of distributed datasets for medical AI training without requiring detailed annotations or centralized data sharing.

## Plain language summary

Breast cancer screening guidelines are expanding to include more MRI scans, increasing the amount of imaging data doctors must analyze. This study explored how artificial intelligence (AI) can help manage this increased workload while overcoming challenges such as limited data sharing between hospitals and the need for detailed annotations on each image. Researchers used MRI scans from five hospitals in the US, Switzerland, the UK, Germany, and Greece to train and test AI models. They found that a specific type of AI model performed the best, and that training AI collaboratively across hospitals improved results compared to training at individual sites. This approach could make AI tools more effective and secure for use in healthcare, potentially improving breast cancer detection and patient outcomes.

New screening recommendations for breast cancer are presently being introduced across Europe and the United States[1,2]. Previous guidelines focused on mammography as the primary tool for breast cancer detection[3,4]. The latest guidelines advocate the use of magnetic resonance imaging (MRI) as a screening method for a significant number of women, particularly those with extremely dense breast tissue. This recommendation is reflected in the recently published EUSOBI guideline[1]. With these modifications, the use of MRI as a screening tool will need to be exponentially scaled up, potentially involving millions of women scanned annually within the European Union. This substantial rise in imaging demand is currently unmatched by a proportional increase in trained specialty radiologists. This disparity underscores a growing medical necessity for computed-assisted approaches, in particular deep learning (DL) methods. These systems can assist radiologists in interpreting breast MRI data, thereby enabling general radiologists to achieve a level of proficiency comparable to that of experts[5]. While high-quality evidence shows a potential clinical benefit of DL in mammography[6], similar advancements in MRI for breast cancer face significant challenges. Notably, studies that have achieved high-performance DL models using MRI data often rely on large, proprietary datasets that are not publicly accessible. The performance of DL systems for medical image analysis scales with the amount of training

✉ e-mail: jakob-nikolas.kather@alumni.dkfz.de

data. Hence, the lack of data accessibility hinders reproducibility and limits collaborative efforts within the research community. Therefore, training accurate, high-performance DL models for breast cancer detection in MRI is constrained by two primary limitations: access to a large number of examinations and the availability of ground truth labels.

Traditionally, DL models for tumor detection in three-dimensional (3D) radiological data are trained in a supervised way, often using manually drawn tumor annotations as a golden standard[7,8]. This process imposes a significant time and labor demand on expert and trained radiologists. Moreover, obtaining precise voxel-level boundaries of tumors in MRI data is not always feasible due to inherent imaging ambiguities and significant intra- and inter-reader variability, which can affect subsequent measurements and model performance. Similar limitations apply to other strongly supervised methods, including bounding-box annotations or centroid annotations, all of which require expert input and can be subjective and ambiguous[9,10]. These factors highlight the importance of alternative approaches that reduce the need for detailed manual annotations. By applying weakly supervised learning using global case labels—readily obtainable from routine radiology reports[11]—we can circumvent the need for precise annotations while mitigating issues related to annotation variability. Studies have shown that automated breast MRI analysis dismissed many lesion free scans without missing any cancer, potentially reducing radiologist workload[12,13].

Training DL models require large and diverse patient datasets, ideally sourced from multiple institutions across different countries. However, sharing such data is complicated by legal, ethical, and privacy concerns, particularly in internal contexts. Federated learning has been proposed as a technical solution to this issue, enabling the training of multiple DL models independently across different sites, thereby countering the need for data sharing[14,15]. Nevertheless, within the context of traditional federated learning, there is a need for a central coordinator to aggregate models from each participating site, introducing a single point of control and potential vulnerability. This centralized aspect can be at odds with the goals of fully decentralized collaboration and may pose scalability issues. Swarm learning (SL) is a more recent advancement which addresses these limitations by using blockchain-based communication and model aggregation between nodes[16,17]. Thus, SL eliminates the need for a central coordinator, allowing participating institutions to contribute equally and securely. Moreover, it allows dynamic participation, which enables seamless onboarding and dropout of participating institutions. While SL has shown promise in previous studies, its application in combination with weakly supervised learning for the analysis of 3D radiological data has not been extensively explored.

In this study, we tackle the dual challenges of the high cost of obtaining ground truth labels and the complexities associated with sharing medical imaging data across institutions. We propose an integrated approach that combines weakly supervised learning with SL to facilitate decentralized training of DL models for breast cancer detection in MRI. To demonstrate the practical feasibility of our approach, we conducted a proof-of-concept study involving three institutions from different countries, deploying on-site hardware to ensure that patient data remained local. We benchmarked a variety of state-of-the-art models—including commonly used two-dimensional (2D) and 3D convolutional neural networks (CNNs), multiple instance learning (MIL)-based models, and vision transformers—within this weakly supervised SL framework. Our results showed that models trained using SL outperformed those trained on local datasets alone, highlighting the potential of SL to enhance model performance through collaborative learning. By providing a practical demonstration of integrating weakly supervised learning with SL for the analysis of 3D radiological data, our study lays the groundwork for larger-scale evaluations. We believe that this approach can significantly advance the development and deployment of DL models in clinical environments, ultimately contributing to improved patient outcomes.

## Methods
### Ethics statement
This study was conducted in accordance with the Declaration of Helsinki. For the University Hospital Aachen (UKA) cohort, ethical approval was granted by the institutional review board (IRB) of UKA (EK 028/19), with data acquired in clinical routine and exported from the local PACS system in anonymized form. The need for informed consent was waived by the IRB, as the data were anonymized. For the Duke cohort[18], ethical approval was granted by the IRB of Duke University Health System, and the anonymized dataset was accessed under the approved protocol; informed consent was not required, as the data are publicly available. For the Cambridge University Hospitals (CAM) cohort, data were collected as part of two clinical trials, TRICKS (REC ref: 13/LO/0411) and BRAID (REC ref: 19/LO/0350), both approved by the Research Ethics Committee of the Health Research Authority and the Integrated Research Application System. Informed consent was obtained from all participants in this dataset. For the Universitaetsspital Zurich (USZ) cohort, ethical approval for this retrospective study was provided by the Ethics Committee of the Canton of Zürich (Kantonale Ethikkommission). Data were accessed from anonymized hospital records under the approved protocol, and the need for informed consent was waived due to the retrospective nature of the study and anonymization of data. Ethical approval for the MHA cohort was granted by the Scientific and Ethics Council of MITERA Hospital, which serves as the hospital's IRB. The council meets monthly to review all projects and research protocols. Access to the MHA dataset was granted exclusively to participants of the ODELIA project, who utilized personalized access passwords to retrieve imaging and clinical data after anonymization. Informed consent was not obtained, as the Scientific and Ethics Council waived the requirement due to the retrospective nature of the study, the anonymization of the data, and the absence of any impact on patient management.

### Patient cohorts
In this retrospective study, we utilized five breast MRI datasets, namely "Duke", "USZ", "CAM", "MHA", and "UKA". For real-world training, Duke, USZ, and CAM were employed as the training cohorts, while UKA and MHA served as the external test cohorts. The training was conducted using real-world SL methodology, which involved training from independent sites without sharing data. The internal training and validation cohort, referred to as "Duke" was hosted on our local system, as it is publicly available[19,20], and was collected between 2000 and 2014 at Duke Hospital in Durham, North Carolina, USA. Out of the 922 patients (=cases) of biopsy-confirmed invasive breast cancer in this dataset, 271 cases lacked information on tumor location and were excluded from analysis. The remaining 651 patients (=cases) were analyzed, consisting of 623 benign and 679 malignant (unilateral) breasts, with 28 patients (=cases) having bilateral breast cancer (Supplementary Fig. 2). As the Duke dataset is a staging dataset all patients had malignant breast cancer. The data was acquired using a 1.5/3.0 Tesla scanner from General Electric or Siemens. The MRI protocol involved a T1-weighted fat-suppressed sequence (one pre-contrast and four post-contrast scans) and a non-fat-suppressed T1-weighted sequence. The study included only female patients, with a mean age of 53 ± 11 years (range 22–90) years. The USZ dataset was employed as a real-world training cohort and was collected between 2013 and 2022 at USZ, Switzerland. The data gathering was conducted within the Picture Archiving and Communication System (PACS) of the USZ. The search focused on identifying dynamic contrast-enhanced (DCE)-MRI examinations that met specific inclusion criteria: individuals aged above 18 years, absence of implants, and availability of assessments for Breast Imaging Reporting and Data System (BI-RADS) indicating the likelihood of malignancy. All the data were acquired in a transverse plane in a prone position and with fat-saturation during the DCE T1 sequences. It includes 272 female patients (Supplementary Fig. 3), out of which 272 were used for analysis, having 203 benign and 69 malignant cases, with 1 of the malignancies representing bilateral breast cancer. The data was acquired using 1.5/3 Tesla scanners from Siemens Sola/Siemens

Magnetom Skyra. The MRI protocol involves a T1-weighted non-fat-suppressed axial sequence (one pre-contrast and four to eight post-contrast scans) and a non-fat-suppressed T2-weighted sequence. The mean age of the patients was $47 \pm 14.5$ years (range 23–83). The CAM dataset was employed as a real-world training cohort and was collected between 2014 and 2021 at CAM, United Kingdom. It includes 305 female patients (Supplementary Fig. 4), out of which 302 were used for analysis, having 153 benign and 149 malignant cases, with 2 of the malignancies representing bilateral breast cancer. The data was acquired using 1.5/3 Tesla scanners from GE SIGNA Artist / DISCOVERY. All the data were acquired in a transverse plane in a prone position and with fat-saturation during the DCE T1-weighted sequences. The UKA dataset[19] was employed as the first external test cohort and was collected between 2010 and 2017 at Aachen Hospital, Germany. It includes 500 female patients (Supplementary Fig. 5), out of which 422 were used for analysis, having 93 benign and 329 malignant cases, with 27 of the malignancies representing bilateral breast cancer. The data was acquired using a 1.5 Tesla scanner from Philips. The MRI protocol involves a T1-weighted, non-fat-suppressed axial sequence (one pre-contrast and four post-contrast scans) and a non-fat-suppressed T2-weighted sequence. The mean age of the patients was $57 \pm 11$ years (range 26–81). The MHA dataset was employed as the second external test cohort and was collected in the year 2022 at Mitera Hospital, Athens. It includes 145 female patients (Supplementary Fig. 6), out of which 144 were used for analysis, having 107 benign and 37 malignant cases, with 5 of the malignancies representing bilateral breast cancer. The data was acquired using a Magnetom Vida 3 Tesla scanner from Siemens. The MRI protocol involves a T1-weighted, non-fat-suppressed axial sequence (one pre-contrast and four post-contrast scans) and a non-fat-suppressed T2-weighted sequence. The mean age of the patients was $50 \pm 11$ years (range 23–81). A detailed description of the image acquisition parameters of all cohorts is reported in (Supplementary Data 1).

### Preprocessing workflow

The same preprocessing pipeline was applied to all datasets in this study. The preprocessing comprises two main steps. In the initial preprocessing step, the DICOM files are converted into NIFTI format, which facilitates the distinction and storage of images as pre-contrast and first post-contrast sequences. Following this, the difference between the first post-contrast and the pre-contrast images is calculated to produce subtraction images, known as sub-contrast sequences. After these initial transformations, individual cropping or padding is applied to the left and right breast volumes to suit the requirements of our model, which processes a single breast volume at a time. We are using intensity-based localization rather than manual segmentation or a separate AI algorithm for segmentation. We crop the height to 256 pixels, It calculates a threshold to find the foreground (presumably the breast area) and adjusts the crop dynamically to include this area, using a margin from the top. Each breast volume is then globally labeled for malignancy (yes/no) according to the classifications provided by the Duke, USZ, CAM, MHA, or UKA datasets. The images are subsequently resampled to achieve a uniform resolution of $256 \times 256 \times 32$ voxels. By simplifying the problem of tumor detection into such a binary classification problem on the whole volume of a breast in an MRI image, we enable the problem to be analyzed with a range of weakly supervised prediction methods (Fig. 1A).

### SL workflow

In this study, we investigate the potential of SL in co-training machine learning models for the purpose of predicting breast cancer on MRI data, utilizing multiple computers that are situated in physically distinct locations. The SL approach enables each participating site to hold its own set of proprietary data, with no clinical data shared between the participants[17]. We implemented an SL network consisting of three separate "nodes", and trained a model using this network (Fig. 1B). During the training process, model weights were exchanged between nodes at multiple synchronization events (sync events), which took place at the end of each synchronization interval. The synchronization interval represents the number of batches

after which learning sharing occurs. The model weights were then averaged at each sync event, and training continued at each node using the averaged parameters. We utilized a weighted averaging approach, which involved multiplying the weights contributed by each node with a weighting factor proportional to the amount contributed by the partner. This approach was motivated by previous studies in gastric and colorectal cancer[16,21]. Our SL implementation stored metadata about the model synchronization on an Ethereum blockchain, with the blockchain managing the global status information about the model. We used the Hewlett Packard Enterprise (HPE) SL implementation, which consisted of four components: the SL process, the Swarm Network (SN) process, identity management, and HPE license management. All processes, or nodes, were run in multiple Docker containers. We provide a detailed description of our SL process, along with a small sample dataset and instructions on how to reproduce our experiments using our code in the following link[22]: https://github.com/KatherLab/swarm-learning-hpe/releases/tag/odelia_v0.6.0.

### Experimental design

In this study, we investigated the performance of multiple DL models within three main categories: CNN-based-2D, CNN-based-3D, and MIL-based workflows (Fig. 1C). We used the SL technique using three nodes, each with a percentage of the training dataset, and compared its performance to the model trained based on a centralized dataset (we will refer to this technique as a centralized model) to validate the efficiency of SL technique as a decentralized learning technique. For this purpose, we used 80% of the Duke dataset for training and the remaining 20% as an internal testing cohort. The UKA dataset was exclusively utilized as an external validation cohort. For the SL technique, we ensured a balanced representation of benign and malignant cases by patient level, dividing the training partition of the Duke dataset with ratios of 40%, 30%, and 10% into three separate learning nodes in three separate bare-metal servers. The data partitioning was conducted randomly at the patient level and in a stratified manner, ensuring that each partition was not based on individual breasts but on the patients as a whole.

To compare the performance of the models trained using the SL technique to the centralized models, we trained each model as a classifier using the centralized training dataset (80% of the Duke dataset) on one single computer system. Following the training phase, all models were subjected to validation on both 20% of the internal cohort and complete external cohorts (UKA). To validate our weakly supervised tumor detection with SL in a real-world scenario, we conducted real-world swarm training. The training involved utilizing Duke's open-source data at our center and the USZ and CAM cohorts at their respective centers. We initiated the models with random weights, a crucial step for introducing variability in the starting conditions for both local and SL setups across participating sites. In local training, each site independently trained a 3D-ResNet101 model on its dataset, allowing models to adapt to unique characteristics of the local data without data exchange between sites. After completing the designated training epochs in the SL framework, a final round of weight merging was performed to align the models across different sites, incorporating features learned from each site's dataset into a unified model. The aggregation algorithm for weights typically averaged the contributions from each participating model. Following the training phase, the models underwent validation against the MHA and the UKA datasets, which were not part of the training data. The experiment was conducted five times to account for variability and to measure the stability of the model's performance across iterations. This approach included repeating the process of weight initialization, model training (both locally and via SL), and the final weight merging step for the SL models. All the experiments were repeated five times using different random seeds.

### AI methods

The DL models chosen for this study were selected through a comprehensive literature survey encompassing radiology and technical publications. Consequently, we explored 2D-CNN models[23–26], 3D-CNN models[27,28], MIL approaches, and also utilized vision transformer (ViT)

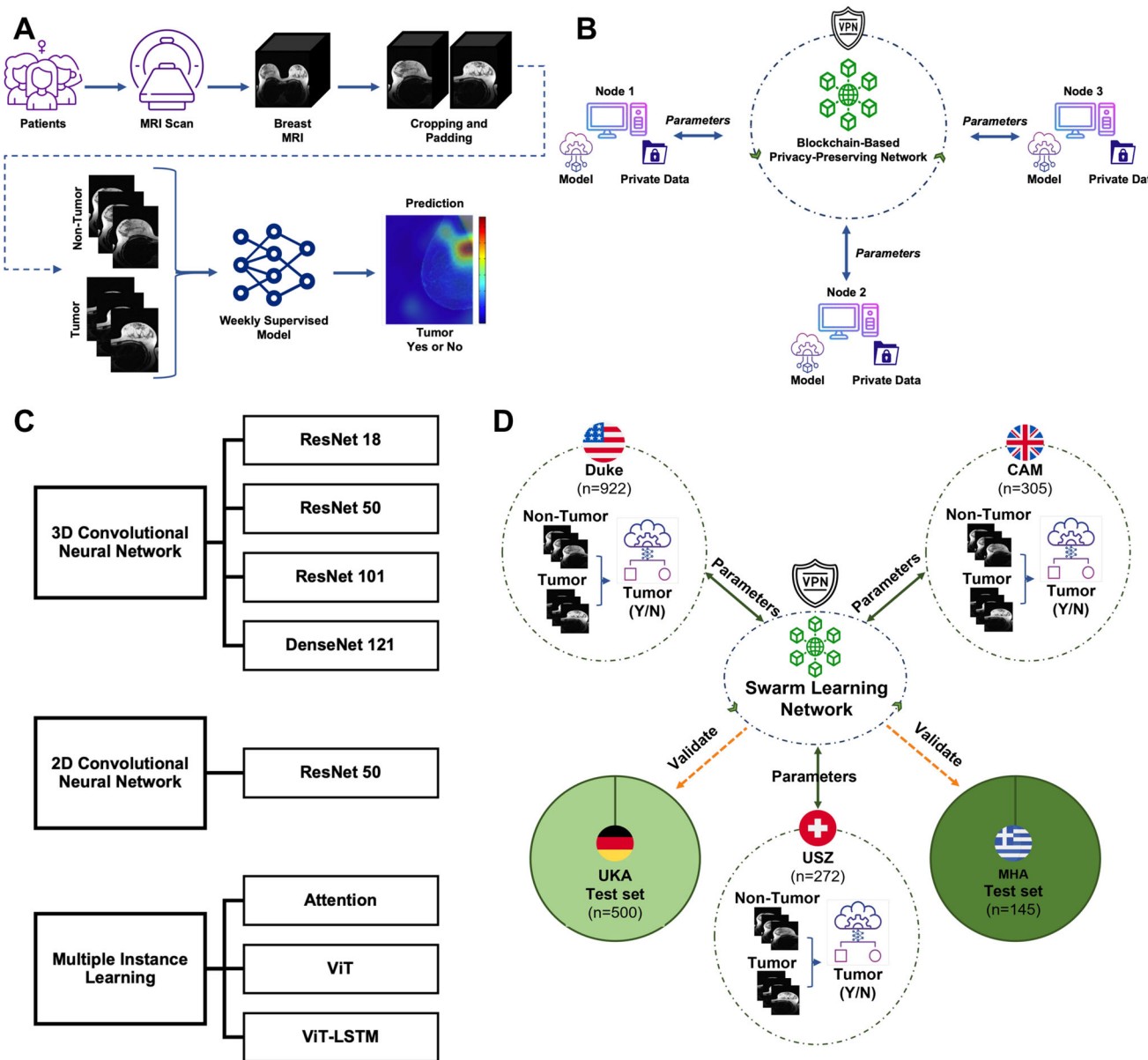

**Fig. 1 | Schematic of the Weakly Supervised Learning (WSL) and Swarm Learning (SL) workflow. A** Schematic representation of the Deep Learning-based WSL workflow for breast cancer tumor detection on Magnetic Resonance Imaging (MRI) data, **B** Overview of the SL setup for a 3-node network, **C** Graphical representation of techniques and models architecture for benchmarking WSL with breast cancer 3D MRI data, **D** Combined representation of real-world SL-based WSL for Breast Cancer Tumor Detection and Data Split Ratio.

models[27,28]. All pipelines were applied to the preprocessed subtraction images as described above. The following provides a detailed description of these techniques.

**2D-CCNs.** CNNs are a subtype of DL models widely used in radiology for activities like image classification, detection, and segmentation. They employ a mathematical function known as convolution to process input images. This makes them capable of learning hierarchical patterns in data in a way that is invariant to translation. To begin with, the 3D MRI data is segmented into 32 slices, each containing $256 \times 256$ pixels. These individual image slices serve as input for CNNs based on the 2D-ResNet50 architecture. Using a weakly supervised learning approach, we label all 32 slices extracted in this scenario based on the volume, which indicates the presence or absence of a tumor in the entire volume. The CNN model treats each slice independently, producing a prediction score that reflects the model's interpretation of specific features or conditions present in each MRI slice. To accumulate individual slice scores and generate volume-level predictions, we selected the highest score among all 32 slices within the volume. This top score was then allocated as the prediction score for the entire volume[29,30].

**3D-CNNs.** These are adaptations of the 2D-CNN design tailored to handle 3D-data for breast MRI classification[31,32]. The 3D-ResNet model incorporates 3D-convolutional layers, pooling layers, and normalization layers along with residual connections, which help retain spatial information and mitigate vanishing gradients during training. By leveraging the 3D structure of breast MRI data, the 3D-ResNet model captures volumetric and spatial information that potentially enhances the classification task's performance relative to 2D-CNN models. In our study, we used 3D-ResNet18, 3D-ResNet50, 3D-ResNet101, and 3D-DenseNet121 architectures. These are adaptations of recognized 2D models specifically designed for volumetric data processing.

**MIL-based methods**. Recent research in developing DL algorithms in the medical domain has identified a promising class of supervised learning algorithms known as MIL[33,34]. Unlike supervised learning, where each observation or "instance" is assigned a separate class label, MIL assigns class labels to groups of observations, or "bags of instances"[35]. MIL operates in two phases: feature extraction (transforming MRI slices into 512 feature vector representations) and training. We used the ResNet18 model pre-trained on ImageNet[36] for feature extraction. Extracted features are then used to train a MIL model to predict outcomes at the slice level. Subsequently, it executes volume-wise aggregation and utilizes an attention mechanism to assess the significance of various instances within a bag for weighing their contributions. This process involves merging predictions from multiple slices that belong to the same volume, resulting in a comprehensive prediction for that MRI volume. We employed three architectures for the MIL-based models: the attention-based MIL[37] (Att-MIL), which has a multilayer perceptron (classifier network) (512 × 256), and (256 × 2) with an attention mechanism[38]. This is followed by a hyperbolic tangent (tanh) layer to obtain the slice-wise prediction score, which is aggregated patient-wise for the 3D-MRI data. The second approach we used is the vision transformer-based MIL[39] (ViT-MIL). In this approach, we used a transformer network with multi-headed self-attention to process slice embeddings. This potentially enables the network to capture intricate relationships among the elements by treating the slice embeddings as a sequential input. The latent dimension of each head is set to 64, resulting in a total dimension of 512. We stack the embeddings of each patch in a sequence of size $n \times 512$, which is then transformed to a dimension of $(n + 1) \times 512$ by applying a linear projection layer followed by ReLU activation. The class token approach was chosen over averaging the sequence elements for better interpretability of the attention heads. The final technique we used for analysis was the ViT-MIL with an LSTM. The architecture starts with a 1D CNN layer to learn local features from the extracted features. It is followed by a linear layer to adapt the input dimensions to the required size for the Transformer Encoder. The Transformer Encoder and BiLSTM layers are then used to model the contextual and temporal relationships within the data. Finally, the attention mechanism is applied to determine the instance-level importance, and the classifier head is used for the final classification. In summary, the ViT-LSTM-MIL model aims to combine the strengths of Vision Transformer, LSTM, and Multi-Instance Learning to improve the classification performance on the 2D slices obtained from the breast MRI dataset.

## Explainability

We utilize three established techniques[40,41] for model visualization: GradCAM[42], GradCAM++[43], and occlusion sensitivity analysis (OCA)[44,45]. GradCAM, or Gradient-weighted Class Activation Mapping, works by visualizing the gradients of the target class in the final convolutional layer of the model. It effectively highlights the regions in the image that have a strong influence on the model's prediction. GradCAM++ is an extension of GradCAM, improving upon its predecessor by using higher-order derivatives and taking pixel-level contribution into consideration, providing a more refined visualization. Occlusion sensitivity works by systematically occluding different parts of the input image and monitoring the effect on the model's output. The change in prediction probability is indicative of the significance of the occluded part in the model's decision. A substantial change implies that the occluded region was critical for the model's decision. The chosen methods operate on a 2D level. Visualization is performed on a selection of 16 slices chosen evenly from the 5th to the 27th slice, ensuring a comprehensive representation of the volumetric data while maintaining a manageable number of visualizations.

## Statistics, reproducibility, and hardware

All experiments were repeated five times with different random seeds. The primary statistical endpoint for classification performance was the area under the receiver operating curve (AUROC). The AUROCs of five training runs (technical repetitions with different random starting values) of a given model were compared. We applied DeLong's test to evaluate and compare the models' performance based on AUROC. To perform the test, we calculated the median patient score from five repetitions of each model. As a result, we determined statistical significance by considering results from DeLong's test with a significance level of $p < 0.05$ as indicative of better performance. AUROCs are reported as mean ± standard deviation. Additionally, we performed more evaluation metrics such as F1 score, sensitivity, specificity, positive predictive value (PPV), and negative predictive value (NPV) on the best-performing model. Sensitivity refers to the ability of the test to accurately identify true positive cases, while specificity measures the ability to accurately identify true negatives. PPV evaluates the probability that a positive test result is indeed a true positive, and NPV assesses the probability that a negative result is accurate.

At three centers, we deployed SL on different hardware configurations for our computational tasks. Duke hosted in Dresden utilized an operating system version of Ubuntu 22.04.3, coupled with 128 GB RAM, and powered by an NVIDIA Quadro RTX 6000 GPU. Meanwhile, USZ operated on Ubuntu 22.04.4, with a more robust setup comprising 256 GB RAM and two NVIDIA GeForce RTX 4090 GPUs. CAM, on the other hand, employed Ubuntu 20.04.6, featuring 62 GB RAM and an NVIDIA RTX 6000 GPU. Furthermore, each system was connected to at least 10 MBit/sec Internet connection, ensuring consistent and reliable network connectivity throughout the study.

## Reporting summary

Further information on research design is available in the Nature Portfolio Reporting Summary linked to this article.

## Results

The first objective of our study was to evaluate whether weakly supervised DL workflows can effectively detect breast cancer in MRI data using only one per-volume label for each patient. To this end, we carried out a systematic comparison of eight distinct weakly supervised prediction workflows, including both 2D- and 3D-CNN models as well as MIL and transformers (Fig. 1A, C). Given that data sharing typically presents a major hurdle in training radiology image analysis pipelines, we hypothesized that SL could alleviate this problem by keeping the dataset distributed throughout different partners. Therefore, we first simulated an SL setup (Fig. 1B) with three nodes set up in one laboratory, each controlling 40%, 30%, and 10% of the data, respectively. Second, we conducted real-world SL experiments, trained on multicentric data, and externally validated them on two test cohorts (Fig. 1D).

### Comparison between simulated SL and centralized mode, using different weakly supervised workflows

In total, we trained the selected models five times using 80% of the Duke dataset and internally validated them on the remaining 20% of the Duke dataset[18], using both SL and centralized models. (Supplementary Data 2) reports the area under the receiver operating characteristic curve (AUROC) values for each experiment for both techniques. Based on the results produced by the SL technique, it was found that 3D-ResNet models performed significantly better than their 2D counterparts (Fig. 2A, Supplementary Table 1, Supplementary Data 4). Notably, among the 3D models, ResNet-101 provided the highest AUROC, reaching 0.792 [±0.045]. The other two 3D-ResNet approaches, 3D-ResNet50 and 3D-ResNet18, also achieved slightly lower performance, 0.766 [±0.050] and 0.768 [±0.022], respectively (Table 1). The MIL techniques in SL, including Vision Transformer-MIL (ViT-MIL), ViT-Long Short Term Memory-MIL (ViT-LSTM-MIL), and Attention-MIL (Att-MIL), showed slightly lower performance with AUROC of 0.740 [±0.019], 0.748 [±0.008] and 0.650 [±0.091] respectively (Table 1). Furthermore, it should be noted that 2D-ResNet50 achieved a significantly lower performance of 0.608 [±0.008] compared to its 3D counterpart ($p = 0.000$). When comparing SL performance using three

**A**    **Duke Data Training on 80% and Validation on 20% for Five Repetitions**

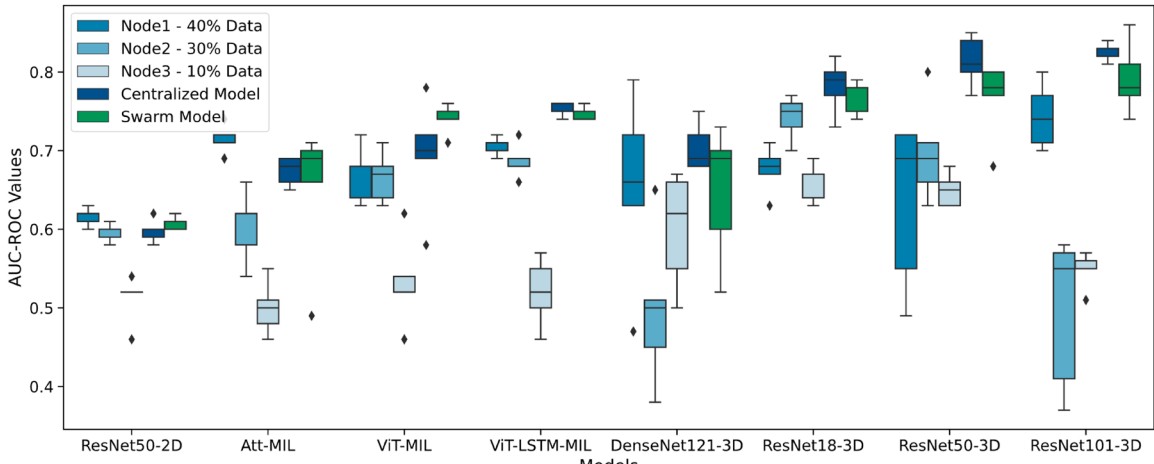

**B**    **Duke Data Training and External Validation on UKA Dataset for Five Repetitions**

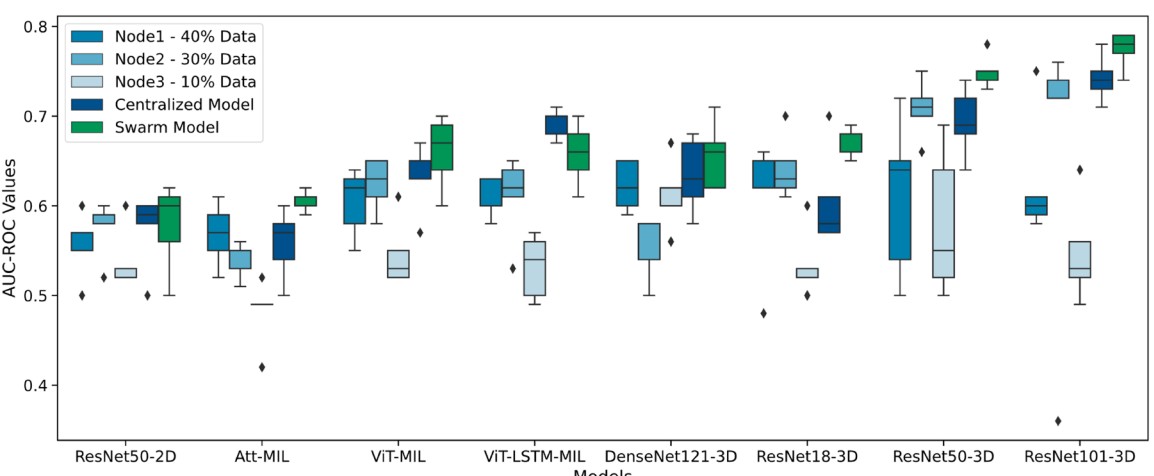

**C**    **Real-world training and External Validation on UKA Dataset**

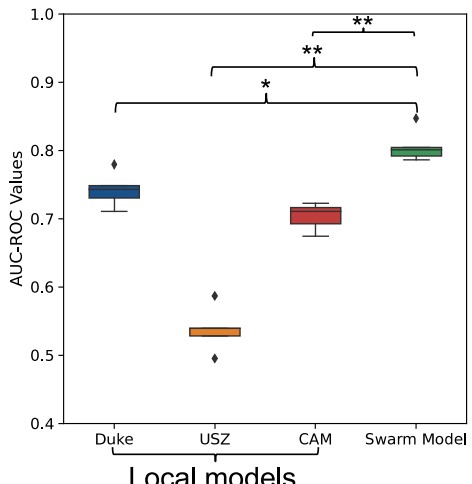

**D**    **Real-world training and External Validation on MHA Dataset**

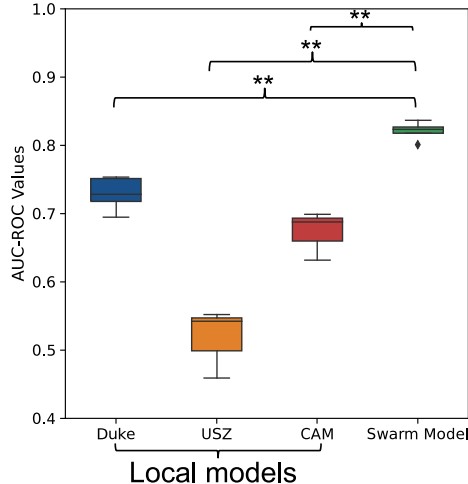

nodes with the performance derived from a centralized model, the SL technique always performed on par with the model trained on the centralized dataset with no significant difference in their performances (Supplementary Tables 2–9).

**Generalizability of different weakly supervised workflows on an external cohort**

To validate our findings in an external cohort and investigate the generalizability of the models between cohorts from multiple origins, we used the

**Fig. 2 | Benchmarking models on internal and external validation.**
**A** Classification performance (area under the receiver operating curve, AUROC) for prediction of tumor on internal validation cohort, i.e., 20% of Duke cohort. The three shades of blue represent different parts of a single cohort, Duke, with the centralized model in dark blue comprising 80% of Duke. Error bars represent the standard deviation of AUROC values for each model across five repetitions of the experiment. Individual data points outside the whiskers indicate outliers from the five repetitions. **B** Classification performance (area under the receiver operating curve, AUROC) for prediction of tumor on external validation cohort, i.e., UKA. The number of patients used for prediction per cohort is 122 for Duke and 422 for UKA. Error bars represent the standard deviation of AUROC values for each model across five repetitions of the experiment. Individual data points outside the whiskers indicate outliers from the five repetitions. **C** Classification performance for prediction of the tumor using 3D-Resnet101 model trained using real-world swarm learning across three cohorts: Duke, USZ, and CAM. Its classification performance was evaluated on an external validation cohort, UKA, for tumor prediction. Local model performance was

assessed using AUROC and DeLong's test to compare it with swarm models. Error bars represent the standard deviation of AUROC values for each model across five repetitions of the experiment. Individual data points outside the whiskers indicate outliers from the five repetitions. The significance level was set at $p < 0.05$ (*$P < 0.05$, **$P < 0.001$), and median patient scores from five repetitions determined superior performance. **D** Classification performance for the prediction of tumors using the 3D-Resnet101 model was trained using real-world swarm learning across three cohorts: Duke, USZ, and CAM. Its classification performance was evaluated on an external validation cohort, MHA, for tumor prediction. Local model performance was assessed using AUROC and DeLong's test to compare it with swarm models. Error bars represent the standard deviation of AUROC values for each model across five repetitions of the experiment. Individual data points outside the whiskers indicate outliers from the five repetitions. The significance level was set at $p < 0.05$ (*$P < 0.05$, **$P < 0.001$), and median patient scores from five repetitions determined superior performance. The training cohort from Duke is consistently represented by the dark blue color throughout the figure.

**Table 1 | Prediction performance statistics for the internal validation of 20% of the Duke cohort. Featuring different nodes and techniques used for benchmarking breast cancer tumor prediction**

| Technique | Local training (AUROC) | | | Swarm (AUROC) | Centralized Model (AUROC) |
|---|---|---|---|---|---|
| | Node1 (40%) | Node2 (20%) | Node3 (10%) | | |
| 3D-ResNet18 | 0.676 [±0.030] | 0.742 [±0.028] | 0.660 [±0.024] | 0.768 [±0.022] | 0.782 [±0.034] |
| 3D-ResNet50 | 0.634 [±0.107] | 0.698 [±0.065] | 0.650 [±0.021] | 0.766 [±0.050] | 0.814 [±0.032] |
| 3D-ResNet101 | 0.744 [±0.042] | 0.496 [±0.098] | 0.550 [±0.023] | **0.792 [±0.045]** | **0.824 [±0.011]** |
| 3D-DenseNet121 | 0.654 [±0.120] | 0.498 [±0.099] | 0.600 [±0.073] | 0.625 [±0.148] | 0.712 [±0.036] |
| ViT-MIL | 0.670 [±0.036] | 0.666 [±0.032] | 0.532 [±0.058] | 0.740 [±0.019] | 0.694 [±0.073] |
| ViT-LSTM-MIL | 0.704 [±0.011] | 0.688 [±0.022] | 0.520 [±0.043] | 0.748 [±0.008] | 0.752 [±0.008] |
| Att-MIL | 0.714 [±0.018] | 0.604 [±0.046] | 0.500 [±0.034] | 0.650 [±0.091] | 0.674 [±0.018] |
| 2D-ResNet50 | 0.614 [±0.011] | 0.594 [±0.011] | 0.512 [±0.030] | 0.608 [±0.008] | 0.598 [±0.015] |

The values of the table represent the mean AUROC, while the errors indicate the standard deviation of AUROC values for each model across five experimental repetitions. The best-performing swarm and centralized models are highlighted in bold.

trained models from internal validation experiments and deployed them on an external dataset from Uniklinik Aachen (UKA), Aachen, Germany.

As in the internal validation experiments, we repeated training and validation of each model five times and reported the AUROC values of each experiment, as well as the mean and median (Supplementary Data 3). With a slight decline in performance for all the models in the external validation experiment we see that again 3D-ResNet models are outperforming other models tested. The performance of 3D-ResNet101 is significantly better than most of the other models for both SL and the centralized data set (Fig. 2B, Table 2, Supplementary Table 10, Supplementary Data 4). 3D-ResNet101 is the highest performing model also in external validation, despite a slight decrease in performance in comparison to the internal validation, reaching an AUROC of 0.770 [±0.021] for the SL technique and 0.742 [±0.026] for the centralized data set. In line with the internal validation results, the validation performance of 2D-ResNet50 using SL on the UKA dataset also reached the lowest performance of 0.578 [±0.049], significantly lower than that of most of the other models (Supplementary Table 10). Comparing the performance of SL and centralized models between internal and external validation revealed that in SL, the performance drop in external validation was much less than in centralized models, indicating that SL models may generalize better than centralized models.

**Real-world training and validation in an international SL network**
To validate our findings in a real-world scenario, we set up an SL training network spanning three institutions in three countries: USZ in Switzerland, CAM in the United Kingdom, and the Duke dataset, residing in Dresden, Germany. We trained a 3D-ResNet101 model architecture across the three sites and validated it in two separate sites, Mitera Hospital Athens (MHA) in

Greece and UKA in Germany. Local models trained on either site were also tested on the UKA and MHA datasets (Tables 3, 4).

We found that on the first external test cohort, UKA, the local models trained on Duke, USZ and CAM achieved AUROCs of 0.743 [±0.025], 0.538 [±0.033], and 0.703 [±0.025] respectively. In comparison, the models trained in the SL setup outperformed all the locally trained models, with an AUROC of 0.807 [±0.024]. The swarm model was significantly better than the local models ($p = 0.035$, 0.001, 0.001, respectively (Fig. 2C, Supplementary Table 11, Supplementary Data 4). The swarm models validated on the UKA cohort achieved an F1 score of 0.624 [±0.029], surpassing the local models at Duke, USZ, and CAM, which attained scores of 0.507 [±0.086], 0.452 [±0.027], and 0.495 [±0.073], respectively.

To investigate the generalizability further, we externally validated all models on a second test dataset from MHA. Here, the local models trained on Duke, USZ, and CAM achieved AUROCs of 0.729 [±0.024], 0.520 [±0.040], and 0.673 [±0.036], respectively. Comparatively, models trained in the SL setup outperformed all the locally trained models, with an AUROC of 0.821 [±0.013]. (Fig. 2D). The swarm model validated on the MHA cohort had an F1 score of 0.596 [±0.036], which is better than the local models. Additional matrices for both local and swarm models, such as sensitivity, specificity, positive predictive value (PPV), and negative predictive value (NPV), are also documented in Tables 3, 4.

Finally, we assessed the explainability of our models' predictions using Gradient-weighted Class Activation Mapping (GradCAM++) and OCA (Fig. 3, Supplementary Fig. 1, Supplementary Methods). In a manual review by a radiologist, we found that GradCAM++ often highlighted irrelevant image regions, and OCA precisely identified malignant enhancing lesions (Fig. 3A, B). These results support the thesis that the models' focus is on tumor areas, with OCA demonstrating more precise localization. Overall,

**Table 2 | Prediction of the performance of different nodes and techniques used for benchmarking breast cancer tumor prediction on the external validation UKA cohort**

| Technique | Local training (AUROC) | | | Swarm (AUROC) | Centralized Model (AUROC) |
|---|---|---|---|---|---|
| | Node1 (40%) | Node2 (20%) | Node3 (10%) | | |
| 3D-ResNet18 | 0.606 [±0.073] | 0.642 [±0.036] | 0.534 [±0.038] | 0.668 [±0.016] | 0.606 [±0.055] |
| 3D-ResNet50 | 0.610 [±0.089] | 0.708 [±0.033] | 0.580 [±0.082] | 0.750 [±0.019] | 0.694 [±0.038] |
| 3D-ResNet101 | 0.626 [±0.070] | 0.664 [±0.171] | 0.548 [±0.057] | **0.774 [±0.021]** | **0.742 [±0.026]** |
| 3D-DenseNet121 | 0.622 [±0.028] | 0.556 [±0.036] | 0.614 [±0.040] | 0.656 [±0.038] | 0.634 [±0.042] |
| ViT-MIL | 0.604 [±0.038] | 0.624 [±0.030] | 0.546 [±0.038] | 0.660 [±0.041] | 0.630 [±0.037] |
| ViT-LSTM-MIL | 0.608 [±0.022] | 0.610 [±0.047] | 0.532 [±0.036] | 0.658 [±0.035] | 0.688 [±0.016] |
| Att-MIL | 0.568 [±0.035] | 0.540 [±0.020] | 0.482 [±0.037] | 0.604 [±0.011] | **0.558 [±0.039]** |
| 2D-ResNet50 | 0.554 [±0.036] | 0.576 [±0.032] | 0.538 [±0.035] | **0.578 [±0.049]** | 0.574 [±0.042] |

The values of the table represent the mean AUROC, while the errors indicate the standard deviation of AUROC values for each model across five experimental repetitions. The best-performing and lowest performing swarm and centralized models are highlighted in bold.

**Table 3 | Prediction performance of different centers used for benchmarking breast cancer tumor prediction on the external validation UKA cohort**

| External Validation on UKA dataset by real-world training with 3D-ResNet101 | | | | |
|---|---|---|---|---|
| | Duke | USZ | CAM | SWARM |
| AUROC | 0.743 [±0.025] | 0.538 [±0.033] | 0.703 [±0.025] | 0.807 [±0.024] |
| F1 Score | 0.507 [±0.086] | 0.452 [±0.027] | 0.495 [±0.073] | 0.624 [±0.029] |
| Sensitivity | 0.635 [±0.144] | 0.430 [±0.06] | 0.589 [±0.078] | 0.767 [±0.021] |
| Specificity | 0.815 [±0.074] | 0.824 [±0.077] | 0.826 [±0.096] | 0.813 [±0.028] |
| PPV | 0.404 [±0.099] | 0.308 [±0.021] | 0.398 [±0.076] | 0.559 [±0.023] |
| NPV | 0.840 [±0.068] | 0.821 [±0.055] | 0.859 [±0.084] | 0.831 [±0.017] |

The table values represent the mean of various evaluation metrics, while the errors indicate the standard deviation of each respective metric for the 3D-ResNet-101 model across five experimental repetitions conducted.

**Table 4 | Prediction performance of different centers used for benchmarking breast cancer tumor prediction on the external validation MHA cohort**

| External Validation on the MHA dataset by real-world training with 3D-ResNet101 | | | | |
|---|---|---|---|---|
| | Duke | USZ | CAM | SWARM |
| AUROC | 0.729 [±0.024] | 0.520 [±0.040] | 0.673 [±0.036] | 0.821 [±0.013] |
| F1 Score | 0.517 [±0.089] | 0.537 [±0.075] | 0.446 [±0.073] | 0.596 [±0.036] |
| Sensitivity | 0.645 [±0.132] | 0.403 [±0.118] | 0.584 [±0.06] | 0.744 [±0.02] |
| Specificity | 0.774 [±0.089] | 0.796 [±0.108] | 0.774 [±0.014] | 0.804 [±0.012] |
| PPV | 0.453 [±0.049] | 0.395 [±0.025] | 0.382 [±0.104] | 0.553 [±0.033] |
| NPV | 0.806 [±0.074] | 0.840 [±0.05] | 0.873 [±0.037] | 0.839 [±0.037] |

The table values represent the mean of various evaluation metrics, while the errors indicate the standard deviation of each respective metric for the 3D-ResNet-101 model across five experimental repetitions conducted.

these data show that in real-world SL experiments, swarm models trained on three cohorts (Duke, USZ, CAM) and validated on external cohorts (UKA, MHA) demonstrated superior performance and better generalizability compared to locally trained models.

## Discussion

Computer-based image analysis of radiology examinations, particularly MRI data, is challenging due to the need for time-consuming and expensive manual annotations of ground truth labels. Additionally, data sharing between medical institutions often faces obstacles due to patient privacy, data ownership, and legal requirements. In this study, we aimed to address these two hurdles by developing DL classification models for radiology. By utilizing weakly supervised DL—which relies on readily available patient

labels instead of detailed manual annotations—we sought to reduce the dependency on strong labels. Furthermore, we incorporated SL to enable collaborative DL model training, as a means to eliminate the necessity for data exchange between collaborating institutions while still benefiting from training on each dataset. Applying this combined strategy to breast MRI datasets, we demonstrated the practical feasibility of integrating weakly supervised learning with SL for cancer detection.

Our findings highlight the practical application of combining weakly supervised learning with SL. Weakly supervised DL allows for efficient processing of radiology images using patient labels that can be semi-automatically generated, reducing the need for extensive manual annotations. While we observed that larger models with more parameters showed better performance even with limited data—a trend consistent with

**Fig. 3 | Visualization of the prediction on external cohorts trained on real-world 3D-ResNet-101 SL model illustrating the findings of our scientific study.** Each row in the visualization corresponds to the best predicted one patient from the external UKA or MHA cohort. The first column displays the center to which the patient belongs. The second column displays 16 slices of the original subtraction images (i.e., the contrast accumulation). The third column shows GradCAM + + visualizations. The last (fourth) column illustrates the results of the occlusion sensitivity analysis (OCA). **A** These are true positive examples. **B** False positive examples. While GradCAM++ highlights regions of the image that are irrelevant to the diagnosis, such as the contrast agent within the heart at the bottom part of the image, OCA focuses on the contrast-enhancing lesions and, thus, on the region that a radiologist would be looking at.

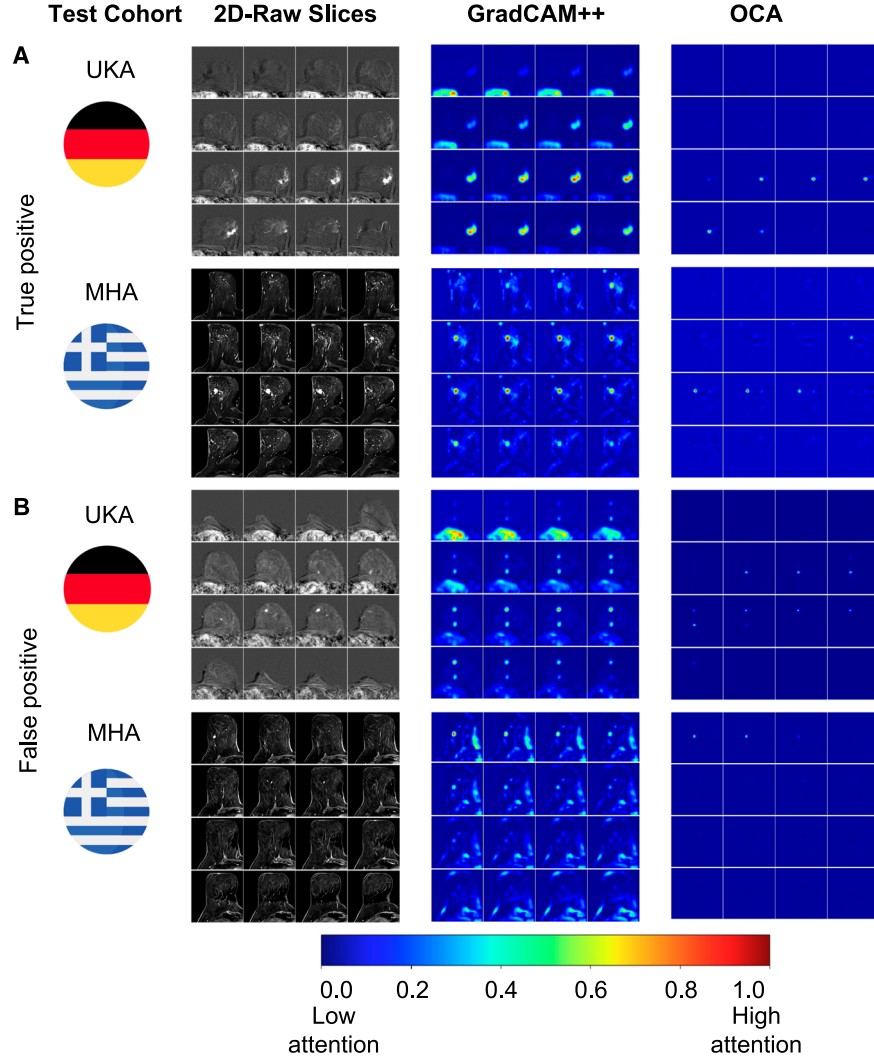

observations in other non-medical domains[46]. We acknowledge that our models did not outperform those trained on larger datasets in previous studies[47]. Concurrently, SL facilitated inter-institutional collaboration without direct data exchange, addressing data privacy and ownership concerns. Although our approach holds promise for large-scale studies spanning multiple hospitals, further research with larger and more diverse datasets is necessary to fully realize this potential.

Our study has several limitations that warrant consideration in future research. A significant limitation is the relatively small size of our training dataset compared to other studies[47–49] that utilized much larger datasets (e.g., over 4000–21,000 patients). This smaller sample size may have impacted our models' performance and limits the generalizability of our findings. Additionally, our models were primarily trained and tested on MRI scans of patients already diagnosed with tumors, which may not reflect the broader screening population where the prevalence of non-malignant findings is higher.

Furthermore, the use of weakly supervised labels—while reducing the need for detailed annotations—may introduce noise and affect model performance, as indicated by the modest AUC scores compared to other studies[47,48]. The three-node SL configuration we employed is relatively simplistic, and real-world collaborations would likely involve more institutions with varying data distributions, introducing additional complexities such as data heterogeneity from different scanners and imaging protocols.

Another limitation is the computational expense associated with higher-dimensional models like 3D-ResNet, potentially posing challenges in

resource-limited environments. For future research, we aim to address these limitations by substantially increasing the patient count and including a broader range of centers worldwide. We acknowledge that other applications, such as chest X-rays and fundus photography, may also be well-suited to these techniques and could be explored in future work. We also plan to investigate strategies to mitigate the impact of noisy labels and improve model performance. While our study provides a proof of concept demonstrating the feasibility of integrating SL with weakly supervised learning, further work is necessary to enhance predictive performance and validate the approach in larger, more diverse populations before it can be considered for developing clinical-grade DL systems in radiology image analysis for MRI-based cancer screening.

## Data availability

The datasets used in this study include Duke, UKA, CAM, USZ, and MHA. The Duke dataset is publicly accessible and can be obtained from The Cancer Imaging Archive (TCIA) via the link: https://doi.org/10.7937/TCIA. e3sv-re93. The UKA, CAM, USZ, and MHA datasets are not publicly available due to privacy and ethical restrictions. These datasets can be accessed upon reasonable request to the authors at the respective sites, subject to approval by the local ethics board and the establishment of a collaboration agreement between the participating institutions. Requests for these datasets should be directed to the authors from the respective centers and will be responded to within 4 weeks. The data for different image acquisition parameters can be found in "Supplementary Data 1" (GR =

Gradient Echo, SE = Spin Echo, DCE = Dynamic Contrast Enhancement). The raw data for prediction performance, measured as AUROC, is provided in "Supplementary Data 2" for experiments conducted with five repetitions using different nodes and techniques on the internal validation 20% Duke cohort. Similarly, the raw data for prediction performance (AUROC) on the external validation UKA cohort, also conducted with five repetitions using different nodes and techniques, is available in "Supplementary Data 3". The source data for Fig. 2 is included in "Supplementary Data 4".

## Code availability

All source code is available at the following link[22]: https://github.com/KatherLab/swarm-learning-hpe/releases/tag/odelia_v0.6.0 The code is based on and requires the Hewlett Packard Enterprise (HPE) implementation of Swarm Learning, which is publicly available at: https://github.com/HewlettPackard/swarm-learning/releases/tag/v2.2.0.

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

## Acknowledgements

The study is organized and funded by the ODELIA consortium, which receives funding from the European Union's Horizon Europe research and innovation program under grant agreement No 101057091. In addition, J.N.K. is supported by the German Federal Ministry of Health (DEEP LIVER, ZMVI1-2520DAT111), the German Cancer Aid (DECADE, 70115166), the German Federal Ministry of Education and Research (PEARL, 01KD2104C; CAMINO, 01EO2101; SWAG, 01KD2215A; TRANSFORM LIVER, 031L0312A; TANGERINE, 01KT2302 through ERA-NET Transcan), the German Academic Exchange Service (SECAI, 57616814), the German Federal Joint Committee (TransplantKI, 01VSF21048) the European Union's Horizon Europe and innovation program (GENIAL, 101096312) and the National Institute for Health and Care Research (NIHR, NIHR213331) Leeds Biomedical Research Centre. The views expressed are those of the author(s) and not necessarily those of the NHS, the NIHR, or the Department of Health and Social Care.

## Author contributions

O.L.S., D.T., and J.N.K. designed the study; O.L.S., J.Z., and G.M.F. developed the software; O.L.S., J.Z., G.M.F, N.R.P., L.E.S., and P.C.V. performed the experiments; O.L.S., J.Z., N.G.L., K.P., M.L., and J.B. analyzed the data; J.Z. performed statistical analyses; S.K., K.A.A., G.A., C.K., A.K., M.K., C.R., S.N., A.A., R.P.L., R.M., W.V., J.C., A.C., C.K., F.J.G. and D.T. provided clinical and radiological MRI data; all authors provided clinical expertise and contributed to the interpretation of the results. V.S., M.W., S.M., D.T., and J.N.K. provided resources and supervision. O.L.S., J.Z., and Z.I.C. wrote the manuscript and all authors corrected the manuscript and collectively made the decision to submit for publication.

## Funding

## Competing interests

J.N.K. declares consulting services for Owkin, France, and Panakeia, UK, and has received honoraria for lectures by Bayer, Eisai, MSD, BMS, Roche, Pfizer, and Fresenius. J.N.K. and D.T. hold shares in StratifAI GmbH, Germany. S.M. declares employment and shareholding with Osimis, Belgium. No other potential conflicts of interest are declared by any of the authors. The authors received advice from the customer support team of Hewlett Packard Enterprise (HPE) when performing this study, but HPE did not have any role in study design, conducting the experiments, interpretation of the results, or decision to submit for publication.

## Additional information

Oliver Lester Saldanha[1,2,20], Jiefu Zhu[1,20], Gustav Müller-Franzes[2], Zunamys I. Carrero [2], Nicholas R. Payne [3], Lorena Escudero Sánchez [3,4], Paul Christophe Varoutas[5], Sreenath Kyathanahally[6,7], Narmin Ghaffari Laleh[1], Kevin Pfeiffer [1], Marta Ligero [1], Jakob Behner[1], Kamarul A. Abdullah[3,8], Georgios Apostolakos[5], Chrysafoula Kolofousi[5], Antri Kleanthous[5], Michail Kalogeropoulos[5], Cristina Rossi[6,7], Sylwia Nowakowska[6], Alexandra Athanasiou [5], Raquel Perez-Lopez [9], Ritse Mann [10,11], Wouter Veldhuis [12], Julia Camps [13], Volkmar Schulz[14,15], Markus Wenzel [14,16], Sergey Morozov [17], Alexander Ciritsis[6], Christiane Kuhl[2], Fiona J. Gilbert [3], Daniel Truhn [2,20] & Jakob Nikolas Kather [1,18,19,20] ✉

[1]Else Kroener Fresenius Center for Digital Health, Medical Faculty Carl Gustav Carus, Technical University Dresden, Dresden, Germany. [2]Department of Diagnostic and Interventional Radiology, University Hospital RWTH Aachen, Aachen, Germany. [3]Department of Radiology, Clinical School, Cambridge Biomedical Research Centre, University of Cambridge, Cambridge, UK. [4]Cancer Research UK Cambridge Centre, Cambridge, UK. [5]Breast Imaging Department, Mitera Hospital Athens, Athens, Greece. [6]Institute of Diagnostic and Interventional Radiology, University Hospital Zurich, Zurich, Switzerland. [7]b-rayZ AG, Schlieren, Switzerland. [8]Universiti Sultan Zainal Abidin, Kuala Nerus, Terengganu, Malaysia. [9]Radiomics Group, Vall d'Hebron Institute of Oncology (VHIO), Barcelona, Spain. [10]Department of Diagnostic Imaging, Radboud University Medical Center, Nijmegen, The Netherlands. [11]Department of radiology, The Netherlands Cancer Institute, Amsterdam, The Netherlands. [12]Imaging Division, University Medical Center Utrecht, Utrecht, The Netherlands. [13]Breast Cancer Unit, Ribera Salud Hospitals, Valencia, Spain. [14]Fraunhofer Institute for Digital Medicine MEVIS, Bremen, Germany. [15]Imaging and Computer Vision, RWTH Aachen University, Aachen, Germany. [16]Constructor University Bremen GmbH, Bremen, Germany. [17]The European Society of Medical Imaging Informatics (EuSoMII), Vienna, Austria. [18]Medical Oncology, National Center for Tumor Diseases (NCT), University Hospital Heidelberg, Heidelberg, Germany. [19]Department of Medicine 1, University Hospital and Faculty of Medicine Carl Gustav Carus, Technische Universität Dresden, Dresden, Germany. [20]These authors contributed equally: Oliver Lester Saldanha, Jiefu Zhu, Daniel Truhn, Jakob Nikolas Kather. ✉e-mail: jakob-nikolas.kather@alumni.dkfz.de

