## [Transparent Peer Review file · Communications Medicine]

Swarm learning with weak supervision enables automatic breast cancer detection in magnetic resonance imaging

Corresponding Author: Professor Jakob Kather

Version 0:

Reviewer comments:

Reviewer #1

(Remarks to the Author)

Given the comments raised in the last round, the authors have added the discussion part on limited data size and novelty claim, I appreciate the authors on acknowledging the limitation and discussing the weakness with potential improvement. Although the authors used MRI breast cancer detection as an example to illustrate the advantage of weak-supervision, there are lots of other applications such as chest x-ray, fundus photo, etc, that might be more suitable for the underlying techniques. The authors might consider trying on these applications or modalities. Overall, I am satisfied with the revision given the scope of Communications Medicine.

Reviewer #3 (Remarks to the Author):

This is the second revision of the manuscript entitled "Swarm learning with weak supervision enables automatic breast cancer detection in magnetic resonance imaging"

The authors chose to add one sentence referencing two publications this reviewer pointed out to the authors in the first evaluation pointing out that there is prior work with sufficiently larger study sizes, better results achieved, published in two different specialty journals in 2021 and 2023, respectively.

At the same time, the authors did not choose to add significantly more patient data outperforming prior work. Instead, the authors argue that "Unfortunately, we cannot use data for training from colleagues in the field because it is not publicly available, and setting up such collaborations would result in several months of delay, which is not feasible for this article." Considering two published studies with clearly more patients included, considering the existence of several radiology networks that should have such data readily available, and considering that the previously published studies show better results, the answer provided is surprising and insufficient. The result of this revision: it takes away large parts of the potential novelty of the presented study, namely showing that large datasets can be generated by bringing many data holders/providers together via the Swarm Learning approach favored by the authors.

Response: We sincerely appreciate your thorough feedback and the opportunity to improve further our manuscript titled "*Swarm Learning with weak supervision enables automatic breast cancer detection in Magnetic Resonance Imaging*".

We acknowledge your concern regarding including a larger dataset and its potential impact on the novelty of our study. Unfortunately, the major obstacle we face remains the availability of external datasets for training, as explained in our previous response. We understand the importance of using larger datasets to demonstrate the power of swarm learning, but accessing such data in the short term remains a challenge due to time constraints and data collection from the participating institutions. While several radiology networks exist additional data collection would not be feasible in a reasonable time frame for the scope of this article. We have addresses this as a limitation in the discussion of the updated manuscript. We have included the line in the discussion section mentioning 'Our study has several limitations that warrant consideration in future research. A significant limitation is the relatively small size of our training dataset compared to other studies^{20–22} that utilized much larger datasets (e.g., over 4,000 to 21,000 patients)'

That being said, we would like to emphasize that our work remains distinct due to its use of weak supervision in combination with swarm learning, which sets it apart from the other published studies. Weak supervision enables us to make effective use of limited labeled data, which is particularly relevant in the medical domain where annotation is time-consuming and expensive. This approach complements swarm learning by demonstrating that even with less granular data, we can achieve competitive performance. We believe this aspect of the study holds significant value and addresses a critical challenge in applying AI to medical imaging.

The authors also claim, “we conducted the first-ever swarm learning-based real-world experiment”. This reviewer would argue that the authors did this already in a wonderful paper published in Nature Medicine in 2021. In other words, the decentralized AI setting is also not the novelty of this manuscript.

As such, this study has potential, but as presented, does not go beyond prior knowledge in the field. In other words, without adding significantly more data, more than in previous studies already published and without illustrating that the addition of significantly more patients' data (than previously already published) would further improve the results provided in previous work, this study falls short of fulfilling important requirements for publication.

Response: Regarding your point on novelty, we appreciate you highlighting our previous work in Nature Medicine (2021). In that study (Warnat-Herresthal et al., Nature Medicine, 2021), swarm learning was applied in a simulated environment, using multiple docker containers on a single server with access to GPUs. Similarly, our Nature Medicine (2022) study (Saldanha et al., Nature Medicine, 2022) employed multiple physical systems within the same institution, simulating a decentralized environment with data from different centers placed on local machines.

In contrast, the current study conducted a real-world swarm learning experiment where data remained on-site across multiple geographically distinct centers, overcoming additional obstacles related to connectivity and setup. This represents a significant difference from the previous studies and is the first of its kind for breast cancer detection using MRI, which presents unique challenges compared to prior applications.

We have revised the manuscript to clarify this distinction better and avoid any misinterpretation of novelty claims. Additionally, we have emphasize the unique contributions of weak supervision in our decentralized learning framework, as well as included a more detailed comparison to the larger-scale studies referenced. We have revised the manuscript and address the concerns raised.

Reviewer #5 (Remarks to the Author): Expert in decentralised machine learning for oncology and cancer imaging

I appreciate the efforts that authors devoted to resolve the concerns from reviewers. Although the authors have added more data, the questions raised by reviewer 3 remain inadequately solved:

1. The samples size, even after the inclusion of new institutes, is relatively smaller compared to existing studies with breast DCE-MRI. For example, in the studies mentioned by Reviewer 3, Verburg et al. included 4,581 patients, and Bhowmik et al. collected 16,535 studies from 8354 patients. In addition, the study by Witowski et al. ("Improving breast cancer diagnostics with deep learning for MRI." Science translational medicine 14.664 (2022): eabo4802.) included 21,537 studies from 13,463 patients. Indeed, the relatively small sample size,

especially the training data (1,499 samples), weakens the demonstration that “large datasets can be generated by bringing many data holder/provider together via the Swarm Learning approach”.

Response: We acknowledge the concern regarding sample size compared to studies like Verburg et al., Bhowmik et al., and Witowski et al. Our study is a proof-of-concept, focused on demonstrating the feasibility of swarm learning in a real-world decentralized setting where data remains on-site. While the sample size is smaller, this reflects the early stages of multi-institutional collaborations and the complexities of decentralized data-sharing. We aim to scale this network over time, and our work shows that swarm learning can achieve meaningful results even with a smaller size of the multicentric dataset. We have mentioned in the discussion ‘We acknowledge that our models did not outperform those trained on larger datasets in previous studies’. We have revised the manuscript and included all the references.

2. This reviewer agrees with Reviewer 3 that the “first-ever swarm learning-based real-world experiment” is not a proper claim of novelty of this paper: the authors did show great potential with their efforts published on Nature Medicine 2021, yet the current study did not show further extension/improvement of the initial contribution in terms of swarm learning. The authors also claimed the novelty of a combination of weakly-supervised learning and swarm learning. However, this results in a combination of separate methods during different stages of the development of the algorithm, and such an integration doesn’t represent a non-trivial solution to a major technology challenge. The advance brought by this study is limited, compared to the authors previous works on swarm learning.

Response: We appreciate your feedback and the opportunity to clarify. While we acknowledge that our previous Nature Medicine 2022 study demonstrated the potential of swarm learning, the current study introduces a real-world, decentralized setting where data remained on-site across multiple institutions this marks a significant logistical and technical challenge that was not addressed in the previous work.

Regarding the combination of weakly-supervised learning and swarm learning, we agree that it involves distinct stages, but we believe this integration offers a practical solution to data scarcity and labeling challenges in medical AI. We have revised the manuscript to refine our claims and more clearly outline the contributions of this study, particularly in addressing real-world constraints of decentralized learning.

3. This reviewer has another concern about the ground truth generated during training. The training labels were obtained using weakly-supervised approaches, which seems brought in massive noise into the models given the relatively low AUC scores (~0.8) obtained even on the Duke internal validation set. In fact, bounding boxes of the breast tumors were provided by the dataset, and it’s not clear why a weakly-supervised approach is necessary here. Given that Duke contains mainly patients with invasive breast cancer, the classification between the invasive lesions and normal slices should be easier: an external AUC of 0.969 on the entire Duke dataset has already been reported by Witowski et al. It is normal that learning from noisy labels usually leads to degraded performance. Nevertheless, such a performance gap reveals that the noisy label problem here represents a more severe

challenge to the whole framework proposed by this work. Further justification or elaboration is required.

Response: We understand your concern about using weakly supervised data. Having detailed labels can indeed improve model performance, but in the real world, these labels aren't always available or easy to get, especially with the partners involved in the study. We chose to use weak supervision to mimic this situation and see how well an AI model could perform without perfect information. While using noisy labels can make the model less accurate, we were able to achieve good results even with this approach. In the abstract we have mentioned 'In this study, we present an integrated pipeline that combines weakly supervised learning—reducing the need for detailed annotations—with local AI model training via swarm learning (SL), which circumvents the need for centralized data sharing, as a way to navigate and address these challenges'. We have made sure to explain this choice more clearly in our paper and talk about the challenges and benefits of using weakly supervised data.

Reviewer #5 (Remarks on code availability):

Although I did not run the experiments personally, a quick look into the codes show the structured codes and documents for reproducing the experiments.

Response: Thank you for your review. We appreciate your time and effort in reviewing our work and are glad to hear that you found our code well-structured and documented to support reproducibility.

Reviewer #1 (Remarks to the Author):

Given the comments raised in the last round, the authors have added the discussion part on limited data size and novelty claim, I appreciate the authors on acknowledging the limitation and discussing the weakness with potential improvement. Although the authors used MRI breast cancer detection as an example to illustrate the advantage of weak-supervision, there are lots of other applications such as chest x-ray, fundus photo, etc, that might be more suitable for the underlying techniques. The authors might consider trying on these applications or modalities. Overall, I am satisfied with the revision given the scope of Communications Medicine

Response: Thank you very much for your thoughtful feedback and suggestions. We appreciate your acknowledgment of the revisions we made regarding data limitations and novelty. We agree that exploring other applications, such as chest X-rays or fundus photography, would be valuable and could further demonstrate the versatility of our approach. We will consider these modalities for future work. We have included a sentence in the discussion section of the paper mentioning 'We acknowledge that other applications, such as chest X-rays and fundus photography, may also be well-suited to these techniques and could be explored in future work'. Thank you again for your insightful review and support.